# Imaging the Binding Between Dasatinib and Its Target Protein in Living Cells Using an SLP Tag System on Intracellular Compartments

**DOI:** 10.3390/ijms26125705

**Published:** 2025-06-13

**Authors:** Da Kyeong Park, Sang-Hee Lee, Hee-Seok Kweon, Zee-Won Lee, Kyung-Bok Lee

**Affiliations:** 1Center for Bio-Imaging & Translational Research and Bioimaging Data Curation Center, Korea Basic Science Institute (KBSI), Cheongju 28119, Republic of Korea; dakyeonge1@kbsi.re.kr (D.K.P.); derbesia@kbsi.re.kr (S.-H.L.); hskweon@kbsi.re.kr (H.-S.K.); 2bHLBIO, Cheongju 28119, Republic of Korea; ezone@bhlbio.com

**Keywords:** drug–target binding, dasatinib, SLP tag system, HaloTag, spatially localized protein expression, subcellular structures

## Abstract

Interactions between chemical drugs and their target proteins are fundamental to drug screening and precision therapy in modern clinical medicine. However, elucidating these interactions within living cells remains challenging due to the limited availability of efficient detection methods. Despite substantial efforts, technical limitations still impede the identification of direct interactors. In this study, we present a simple method to detect the binding between a chemical drug and its target proteins in live cells. This approach utilizes a self-labeling protein (SLP) tag system, specifically HaloTag which is a modified haloalkane dehalogenase, combined with spatially localized expression of the SLP. To implement this system, dasatinib was conjugated to a HaloTag ligand, and the HaloTag protein was expressed in specific intracellular compartments, such as endosomes or F-actin structures. Upon treatment of cells with the HaloTag ligand-conjugated dasatinib, green fluorescent protein (GFP)-fused cytoplasmic dasatinib target proteins were observed to co-localize with the HaloTag at these subcellular structures, thereby indicating direct drug–target binding. This method provides a good spatial resolution with a high signal-to-noise ratio and low false-positive signals across a high background and false-positive/false-negative signals from endogenous proteins and/or non-specific binding. In this context, we believe that our method is a useful platform for visualizing the binding between chemical drugs and their cytoplasmic targets within living systems.

## 1. Introduction

A thorough understanding of how chemical drugs bind to their target proteins in live cells is crucial for designing therapeutic agents with improved selectivity and specificity. This critical interaction—an essential prerequisite for therapeutic efficacy—is often poorly understood [1]. In live cells or whole organisms, this interaction generally cannot be visualized due to a lack of methods for directly measuring drug–target engagement in situ. Consequently, our current understanding remains incomplete and relies on target extraction assays or indirect measurements that lose critical spatiotemporal information, thereby complicating drug development [2]. A variety of techniques have been developed to study drug–target interactions in living cells, including patch–clamp recording, dark-field optical microscopy, fluorescence spectroscopy, and imaging approaches such as total internal reflection fluorescence (TIRF), fluorescence polarization (FP), fluorescence resonance energy transfer (FRET), fluorescence correlation spectroscopy (FCS), and super-resolution microscopy (SRM) [3,4]. These techniques allow for the investigation of subcellular pharmacokinetics and dynamics, providing detailed spatial and temporal data on drug–target interactions in live cells. However, fluorescent labeling of drugs can significantly alter their pharmacological activity [5]. Therefore, accurately measuring intracellular interactions between unlabeled drugs and their targets under physiological conditions remains an urgent challenge. 

Imaging chemical drug–target proteins expressed in the cytoplasm has been challenging due to high background noise, false-positive/false-negative signals from endogenous proteins, and non-specific cytoplasmic binding [6]. To solve these problems, we envisioned that the spatially localized expression of target proteins in specific intracellular compartments of a cell would simplify the readout of the binding between chemical drugs and their target proteins. Redistribution and co-localization assays employing specific intracellular compartments as recruitment/redistribution platforms are widely used to monitor molecular interactions inside cells [7,8,9,10]. In a previous study, we developed a simple method to visualize the binding between a chemical drug and its target proteins in the cytoplasm using a two-step bioorthogonal labeling strategy combined with spatially localized protein expression [11]. Dasatinib was modified with *trans*-cyclooctene (TCO), yielding dasatinib−TCO, and its cytoplasmic target kinases were expressed in specific intracellular compartments, such as endosomes and F-actins. After bioorthogonal labeling, co-localization between dasatinib and its target proteins was observed in these compartments. To express dasatinib–target kinases at specific subcellular structures, we prepared fusion constructs such as Rab5b−mCherry−kinase for endosome and LifeAct−TagRFP−kinase for F-actin, which were transiently expressed in HeLa cells. The cells were first incubated with dasatinib−TCO in a serum-free medium. After the removal of the excess probe, CFDA−Tz (carboxyfluorescein diacetate–tetrazine) was added to the growth medium. Co-localization between dasatinib−TCO and CFDA−Tz was observed in endosomes and F-actins.

In this study, we expanded the capabilities of our spatially localized protein expression assay to visualize interactions between a chemical drug and its target proteins in live cells. We designed a fusion construct which is capable of anchoring chemical drugs and localizing expression to specific subcellular structures, thereby potentially guiding the co-localization of cytoplasmic target proteins to defined intracellular compartments upon treatment with a self-labeling protein (SLP) tag ligand-conjugated chemical drug. This approach may effectively reduce non-specific interactions between drugs and proteins, enabling the monitoring of chemical drug–protein interactions in live cells (Figure 1).

## 2. Results and Discussion

Self-labeling protein (SLP) tags enable the covalent labeling of fusion proteins with synthetic molecules, facilitating various applications in bioimaging and biotechnology [12]. In this approach, nearly any probe that can be functionalized with an appropriate linker can be conjugated to its respective tag and used to visualize or manipulate the function and dynamics of a fused protein of interest (POI) [13]. One such SPL is the HaloTag protein (HTP), a modified haloalkane dehalogenase engineered to covalently bind synthetic ligands known as HaloTag ligands (HT ligands). These ligands consist of a chloroalkane linker attached to various functional molecules, such as fluorescent dyes, affinity tags, or solid supports. The covalent bond formation between HTP and the HT ligand is highly specific, occurs rapidly under physiological conditions, and is essentially irreversible [14]. The HaloTag system is a versatile protein-labeling technology used in a wide range of applications, including protein purification, the analysis of protein−protein and protein−DNA interactions, and studies of protein localization. The system functions by genetically fusing HTP to a PIO and labeling or manipulating specific HT ligands [15]. 

In this work, we employed the HaloTag system to visualize interactions between a chemical drug and its target proteins within specific intracellular compartments, such as endosomes and F-actins. Briefly, HTP was expressed on endosomes and F-actins, while GFP-fused dasatinib–target kinases were expressed in the cytoplasm. Upon treatment with HT ligand-conjugated dasatinib (dasatinib−HT ligand), we imaged the interactions between dasatinib and its target kinases on the endosome and F-actin. Dasatinib (Sprycel; Bristol-Myers Squibb) is a small-molecule inhibitor of BCR-ABL and SRC-family tyrosine kinases (SFKs). It is approved for the treatment of patients with imatinib (Gleevec; Novartis)-resistant chronic myelogenous leukemia (CML) and Philadelphia chromosome-positive acute lymphoblastic leukemia (Ph+ ALL) [16]. ABL and SFKs are a group of non-receptor tyrosine kinases that are widely expressed in the cytoplasm [11]. Owing to their widespread expression in the cytoplasm, dasatinib was chosen as the model drug.

To verify our approach, we synthesized dasatinib−HT ligand via a two-step reaction (Figure 2A and Appendix A). The terminal hydroxyl group of dasatinib was converted to an amino group, which was subsequently coupled with the HaloTag−succinimidyl ester (O2) ligand. Several previous studies have shown that modifications to the terminal hydroxyl group of dasatinib do not significantly affect its binding affinity to target proteins [11,17,18,19,20].

To confirm that conjugation with the HT−ligand did not significantly change the binding affinity to its target proteins, we first evaluated the inhibitory effect of dasatinib−HT ligand with recombinant active kinase ABL1 using a standard Kinase-Glo Plus luminescence assay. The observed IC_50_ value of dasatinib−HT ligand under the employed conditions was 152.8 nM, which was approximately 7-fold higher than that of dasatinib itself (22.12 nM) (Figure 2B). Next, to investigate the inhibition of autophosphorylation at Tyr416 of SRC in cells, full-length SRC (EGFP-SRC) was transiently expressed in HeLa cells, followed by Western blot analysis. Upon treatment with different doses of dasatinib−HT ligand (1 and 5 μM) for 1 h, the levels of active, phosphorylated SRC significantly decreased compared with those in vehicle-treated HeLa cells (Figure 2C). Taken together, these results indicated that the dasatinib−HT ligand was cell-permeable and a good mimic of dasatinib, with the ability to bind to its targets in live cells.

To express HTP in intracellular compartments such as endosomes and F-actins, we prepared the following red fluorescent fusion proteins: Rab5b−mCherry−HTP and LifeAct−TagRFP−HTP, respectively (Appendix A) [7,11]. Rab5b is a member of the Rab family of small G-proteins, and its overexpression in cells leads to the formation of enlarged endosomal vesicles [7]. LifeAct is a 17-amino acid peptide that specifically binds to F-actin without interfering with actin dynamics in vitro and in vivo. It is therefore widely used as a molecular probe to visualize actin dynamics in both live and fixed cells when genetically fused to fluorescent proteins [21]. After preparation of the fusion constructs, Rab5b−mCherry−HTP and LifeAct−TagRFP−HTP were transiently expressed in HeLa cells. Expression of the HTP was visualized in cells, followed by staining with the HT−mAb and HT−diAcFAM ligand (cell-permeable green fluorescent dye, a quick and simple way to label expressed HaloTag fusions within live cells) [13]. As expected, HTP was expressed on the surface of the intracellular compartments (Figure 2D).

After confirming the cellular uptake and affinity of dasatinib−HT ligand and verifying spatially localized HTP expression, we obtained images of its binding to cytoplasmic target kinases in live cells. The target kinases of dasatinib (ABL1, BLK, BMX, BTK, CSK, FGR, HCK, LYN, SRC, and YES1) were cloned into an EGFP vector (Appendix A) [22]. Rab5b−mCherry−HTP (or LifeAct−TagRFP−HTP), and each GFP−fusion kinase was transiently co-expressed in HeLa cells. HTP was expressed in intracellular compartments, such as endosomes and F-actins, while the GFP-fusion kinase was expressed in the cytoplasm (Figure 3, left panel, and Appendix A, left panels). The cells were treated with dasatinib−HT ligand for 1 h in a serum-free medium (final concentration: 1 M, 0.1% DMSO). As shown in Figure 3 (right panel), the GFP-fusion kinase co-localized with the intracellular compartments (Appendix A, right panels). This assay employs two distinct protein constructs: a bait protein localized to either endosomes or F-actin via an SLP tag and a cytosolic prey protein consisting of a GFP-fused kinase. Upon treatment with the HT ligand-conjugated chemical drug, the GFP−fusion kinase is recruited to endosomes (or F-actin), indicating an interaction between the chemical drug and its target proteins. Successful recruitment is visualized by the appearance of green circles (for endosomes) and lines (for F-actin), reflecting the specific localization of the bait protein. For a negative control experiment, GFP and Rab5b−mCherry−HTP (or LifeAct−TagRFP−HTP) were co-expressed in HeLa cells (Appendix A, left panel). The cells were treated with dasatinib−HT ligand (1 µM) for 1 h in a serum-free medium. GFP was not co-localized with the intracellular compartments (Appendix A, right panel), indicating the absence of nonspecific interactions between the dasatinib and GFP. To further validate the specificity of the dasatinib−HT ligand binding, competition assays were performed using excess dasatinib. The binding signal (green circles (on endosomes) and lines (on F-actin)) between dasatinib−HT ligand and GFP−ABL1 was decreased substantially by brief co-incubation with an excess of dasatinib (10 µM) in the same cells (Figure 4). Overall, these results clearly show that our assay can be used to visualize the interaction between a chemical drug and its target proteins in living cells using a GFP-labeled protein library.

## 3. Materials and Methods

### 3.1. Synthesis of HT Ligand-Conjugated Dasatinib (Dasatinib−HT Ligand)

All reagents were obtained from commercial sources and used without further purification. Dasatinib was purchased from LC Laboratories (Woburn, MA, USA). The HaloTag−succinimidyl ester (O2) ligand was purchased from Promega (Madison, WI, USA).

Amine-modified dasatinib was prepared, beginning with commercially available dasatinib, according to the literature method [11]. The HaloTag−succinimidyl ester (O2) ligand (129.4 mg, 0.22 mmol) was added to a solution of amine-modified dasatinib (100 mg, 0.21 mmol) in anhydrous DMF (1.5 mL). The mixture was stirred for 12 h, and the solvent was removed under reduced pressure. The crude residue was purified via silica gel flash column chromatography (DCM:MeOH = 7:1 to 4:1) to yield the desired product, dasatinib−HT ligand (142.1 mg, 72%), as a white solid.

### 3.2. In Vitro Kinase Assay

The IC50 of dasatinib−HT ligand for ABL1 was determined using the Kinase-Glo® Plus luminescent assay kit (Promega, Madison, WI, USA). The recombinant active kinase ABL1 and a synthetic peptide substrate (ABL1: EAIYAAPFAKKK) were purchased from SignalChem (Richmond, BC, Canada). The ATP and substrate peptide concentrations used in the assay were 100 μM and 100 μM, respectively. Dose-dependent inhibition assays were performed by varying the concentration of the probe under a kinase concentration of ~25 nM. The IC50 values of the probes were calculated from percentage activity vs. log [concentration of probe] curves generated using the GraphPad Prism 10 software.

### 3.3. SRC Kinase Inhibition Assay in Cells

HeLa cells (ATCC, Manassas, VA, USA) were transiently transfected with EGFP−SRC. The transfected cells were washed with serum-free DMEM. The cells were treated with either dasatinib or dasatinib−HT ligand (1 and 5 M, 0.1% DMSO in serum-free DMEM) for 1 h. They were then washed with PBS, harvested using a cell scraper, and collected via centrifugation (16,000× *g* for 20 min. in a 4 °C precooled centrifuge). Immunoblotting analysis was performed with pY416-SRC (1:1000; #2101) and SRC (#2108) antibodies (Cell Signaling Technology, Danvers, MA, USA).

### 3.4. Construction of Rab5b−mCherry−HTP and LifeAct−TagRFP−HTP Expression Vectors

The Rab5b^Q79L^−mCherry−FRB vector was kindly provided by Professor W. D. Heo (KAIST, Daejeon, Republic of Korea). The LifeAct−TagRFP vector was purchased from ibidi (Gräfelfing, Germany).

To construct the Rab5b−mCherry−HTP expression vector, the FRB gene was removed from the Rab5b−mCherry−FRB using the restriction enzymes NheI and AgeI. The PCR-amplified HTP gene was cloned into the FRB gene-removed Rab5b−mCherry−FRB vector.

To construct the LifeAct−TagRFP−HTP expression vector, the EGFP gene was removed from the EGFP−C3 vector (Clontech Inc., San Jose, CA, USA). The PCR-amplified LifeAct−TagRFP gene was cloned into the EGFP gene-removed EGFP−C3 vector. After constructing the LifeAct−TagRFP−C3 vector, the PCR-amplified HTP gene was cloned into the LifeAct−TagRFP−C3 vector. The HaloTag (HT) gene was amplified from the HT−Pit1 plasmid (Addgene, Watertown, MA, USA) via PCR, using primers. The primers and vectors used in this study are listed in Supporting Appendix A.

### 3.5. Construction of EGFP−Kinase Expression Vectors

The ABL1 (BC117451) gene was obtained from Addgene (plasmid #23939, Watertown, MA, USA). The wild-type genes BLK (NM_001715), BMX (NM_001721), BTK (NM_000061), CSK (BC106073), FGR (NM_005248), HCK (BC014435), LYN (NM_001111097), SRC (BC011566), and YES1 (NM_005433) were obtained from the Korean Human Gene Bank (KHGB, Daejeon, Republic of Korea) and Open Biosystems (Thermo Fisher Scientific Inc., Waltham, MA, USA). To construct the EGFP−kinase expression vectors, each of the PCR-amplified kinase genes was cloned into either the EGFP−C3 or EGFP−N1 vector (Clontech Inc., San Jose, CA, USA). The primers and vectors used in this study are listed in Supporting Appendix A.

### 3.6. Validation of HaloTag−Fusion Proteins

For immunofluorescence imaging, HeLa cells were grown on an 8-well Nunc Lab-Tek II chambered coverglass (Thermo Fisher Scientific Inc., Waltham, MA, USA) and transfected with Rab5b−mCherry−HTP or LifeAct−TagRFP−HTP. The cells were rinsed twice with PBS, fixed with 4% paraformaldehyde in PBS for 5 min, and permeabilized with 0.1% Triton X-100 in PBS for 10 min. The cells were then washed with PBS three times and incubated with 2% normal goat serum and 0.01% Triton X-100 in PBS overnight at 4 °C. The cells were incubated with primary anti-HaloTag monoclonal antibody (1:1000, #G9211, Promega, Madison, WI, USA) for 2 h at room temperature, and the Alexa Fluor (488)-conjugated anti-rabbit IgG (1:2000, A-11008, Invitrogen, Waltham, MA, USA) was used to reveal antigen−antibody complexes via incubation for 45 min at room temperature. After washing three times with PBS, the samples were used for confocal imaging.

For fluorescence imaging, the HaloTag−diAcFAM ligand (HT−diAcFAM ligand) (Promega, Madison, WI, USA) was used to label cells expressing Rab5b−mCherry−HTP or LifeAct−TagRFP−HTP, following the manufacturer’s instructions with some modifications. In brief, HeLa cells were grown to 50–70% confluence on 25 mm round coverslips (Paul Marienfeld GmbH & Co. KG, Lauda-Königshofen, Germany) in a 6-well cell culture plate. Transient transfection of Rab5b−mCherry−HTP or LifeAct−TagRFP−HTP was conducted using a TurboFect (Thermo Fisher Scientific Inc., Waltham, MA, USA) according to the manufacturer’s standard protocol. The transfected cells were rinsed once with serum-free DMEM and treated with HT−diAcFAM ligand (5 M in serum-free DMEM). After 15 min, the cells were rinsed twice with PBS and incubated in fresh culture medium for 30 min. The medium was then replaced with fresh medium or PBS, and the samples were used for confocal imaging.

For the Western blot experiment, HeLa cells were grown on a cell culture plate to 50–70% confluence. The cells were transfected with Rab5b−mCherry−HTP or LifeAct−TagRFP−HTP, washed twice with cold PBS, harvested using a cell scraper, and collected via centrifugation (16,000× *g* for 20 min. in a 4 °C precooled centrifuge). The cell pellets were washed with PBS, lysed with a pro-prep protein extraction solution (Intron biotech., Seongnam, Republic of Korea), and then incubated on ice for 1 h. The lysates were centrifuged, and the supernatants were collected. The total protein concentration was determined using the Bradford assay procedure. A total of 20 g of each protein sample was boiled for 10 min, separated via electrophoresis on 12% polyacrylamide gel, and electro-transferred to a nitrocellulose membrane. Nonspecific binding sites were blocked in Tris-buffered saline (TBS) containing 5% BSA and 0.1% Tween-20 at room temperature for 1 h. The membranes were rinsed and then incubated with anti-HaloTag monoclonal antibody (1:1000) at 4 °C overnight, followed by horseradish peroxidase-conjugated goat anti-rabbit IgG (1:2000, sc-2004, Santa Cruz Biotechnology, Dallas, TX, USA) at room temperature for 2 h. After the immunocomplexes were rinsed with a buffer, they were visualized with chemiluminescence using the ImageQuant LAS 4000 (GE Healthcare Bio-Science Corp., Piscataway, NJ, USA), according to the manufacturer’s instructions.

### 3.7. Confocal Imaging

HeLa cells were grown to 50–70% confluence on 25 mm round coverslips in a 6-well cell culture plate. Transient co-transfection of the desired protein pair (Rab5b−mCherry−HTP/EGFP−ABL1(SRC, CSK, etc.) and LifeAct−TagRFP−HTP/EGFP−ABL1(SRC, CSK, etc.)) was conducted using TurboFect. The co-transfected cells were rinsed once with serum-free DMEM, and the cells on round coverslips were mounted on a homemade perfusion chamber connected to a temperature controller set to 37 °C. The cells were treated with 1M dasatinib−HT ligand (0.1% DMSO in serum-free DMEM) for 1 h. Live cells were imaged using a laser-scanning confocal microscope (LSM 800 with Airyscan, Carl Zeiss, Oberkochen, Germany) (OC101) with a C-Apochromat 40×/1.2 water immersion lens at the Korea Basic Science Institute (Ochang, Republic of Korea).

## 4. Conclusions

In this work, we demonstrated that the binding between a chemical drug and its target proteins in the cytoplasmic region can be visualized in living cells by combining an SPL tag with spatially localized protein expression. To validate our approach, we designed fusion constructs, Rab5b–mCherry–HTP and LifeAct–TagRFP–HTP, that are capable of both anchoring chemical drugs (an HT ligand-conjugated chemical drug) and localizing protein expression to intercellular compartments, such as endosomes and F-actin. We first synthesized the dasatinib−HT ligand. Rab5b−mCherry−HTP (or LifeAct−TagRFP−HTP) and GFP-fused dasatinib–target kinases were transiently co-expressed in HeLa cells. Imaging of the cells after treatment with dasatinib−HT ligand clearly showed binding events between dasatinib and its target kinases (ABL1, SRC, CSK, etc.) at specific subcellular structures. We believe that this method provides a useful platform for visualizing the binding between chemical drugs and their cytoplasmic targets within living systems. We are currently exploring additional proteins and small molecules in protein–drug binding studies.

## Figures and Tables

**Figure 1 ijms-26-05705-f001:**
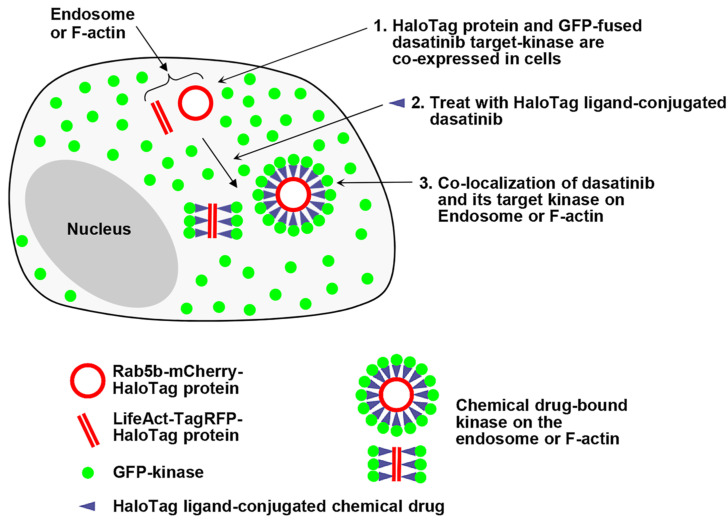
A schematic illustration of the strategy: Live cell monitoring of the chemical drug binding to its target protein is performed within spatially localized intercellular compartments, such as endosomes or F-actin. (1) HeLa cells are co-expressed with a HaloTag fusion protein localized to the surface of endosomes (or F-actin) and with a green fluorescent protein (GFP)-fused dasatinib–target kinases. (2) These cells are then treated with a HaloTag ligand-conjugated dasatinib, and (3) the co-localization of the chemical drug and its target kinases is observed in endosomes (or F-actin).

**Figure 2 ijms-26-05705-f002:**
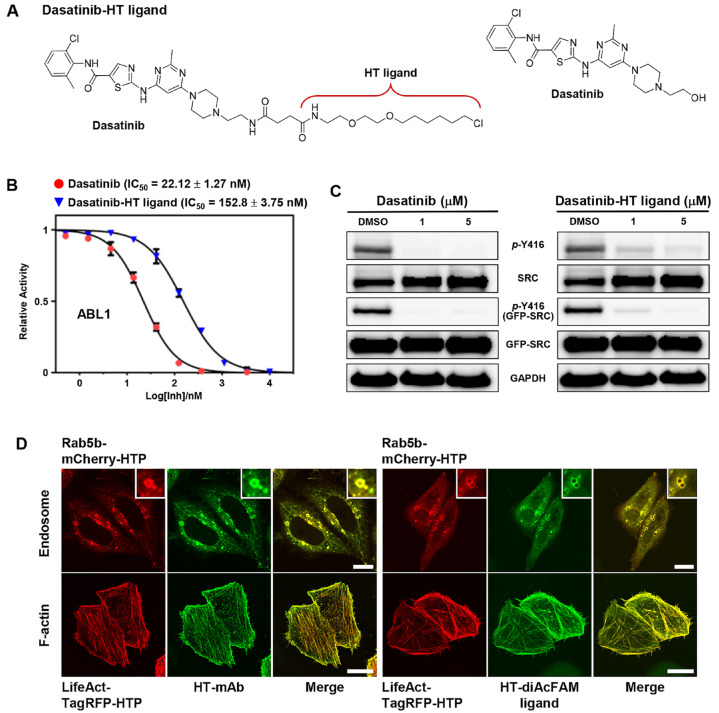
(**A**) Chemical structure of dasatinib−HT ligand. (**B**) IC_50_ plots for dasatinib and dasatinib−HT ligand against recombinant ABL1 kinase. (**C**) Western blot analysis showing the inhibition of autophosphorylation of Y416 of SRC transiently expressed in HeLa cells with dasatinib and dasatinib−HT ligand. (**D**) Confocal images of endosome-localized Rab5b−mCHerry−HTP fusion protein (top) and F-actin-localized LifeAct−TagRFP−HTP fusion protein (bottom), labeled with HT−mAb and HT−diAcFAM ligand, respectively. All scale bars are 10 μm.

**Figure 3 ijms-26-05705-f003:**
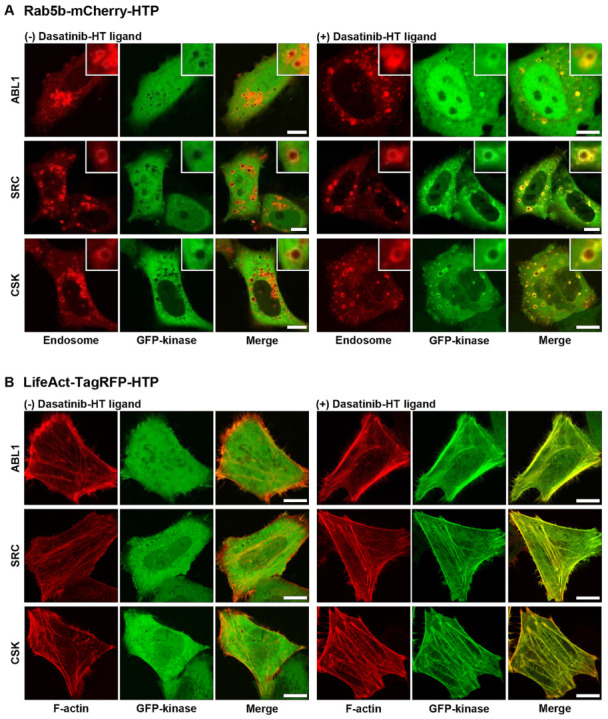
Binding between dasatinib and its target proteins on endosome (**A**) and F-actin (**B**) in untreated (left) and dasatinib−HT ligand-treated (right) cells. All scale bars are 10 μm.

**Figure 4 ijms-26-05705-f004:**
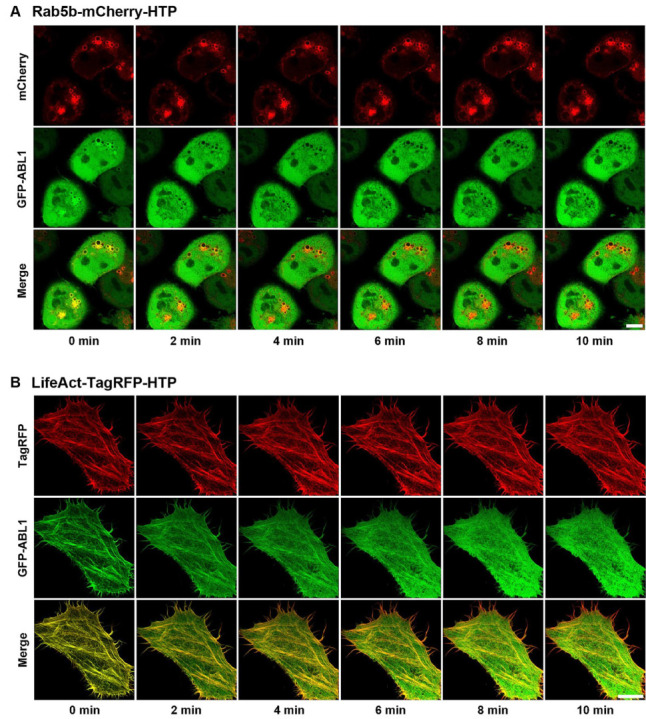
Dasatinib-mediated inhibition of binding between dasatinib−HT ligand and GFP−ABL1 on the endosome (**A**) and F-actin (**B**) in the same cells. All scale bars are 10 μm.

## Data Availability

Data are contained within the article and in the Appendix A.

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
