# Peer review of "Imaging the Binding Between Dasatinib and Its Target Protein in Living Cells Using an SLP Tag System on Intracellular Compartments"

_ijms, 2025, doi:10.3390/ijms26125705_

Round 1
Reviewer 1 Report
Comments and Suggestions for Authors
In this study, Park et al. present a novel workflow to monitor drug–protein interactions in live cells. Using HaloTag protein, a chloroalkane-tagged version of dasatinib, and GFP-tagged kinases as a model system. They visualize drug-target engagement within specific subcellular compartments (endosome, F-actin) via confocal microscopy. The manuscript includes a clearly described methods section, and the topic is relevant to the journal’s readership. However, several key issues should be addressed before the manuscript can be considered for publication:
- Please replace the current cartoon in Figure 1 with a more detailed, step-by-step schematic to more clearly convey the experimental workflow.
- The Results and Discussion section lacks adequate interpretation and contextualization. The authors should expand on the results, provide deeper insights into their implications, and relate the findings to prior literature—particularly to make the study more accessible to readers unfamiliar with confocal microscopy.
- The manuscript does not address the potential for false-positive interactions using this strategy. To strengthen the study, I encourage the authors to include a negative control: for example, GFP-tagging a protein known not to interact with dasatinib and repeating the imaging experiment to assess background binding.
Author Response
"Please see the attachment"

Reviewer 2 Report
Comments and Suggestions for Authors
This manuscript reports the study of using a HaloTag ligand-conjugated dasatinib to image the direct interactions between GFP-fused dasatinib target kinases and HaloTag-fused proteins within endosomes and F-actins. The results clearly show that HaloTag ligand-conjugated dasatinib serves a good probe for imaging spatially localized protein expression. The manuscript is accepted for publication after addressing a few minor issues.
Comments:
#1: In the caption of Figure 1, it is recommended to use the full terms for HTP and HT.
#2: High-resolution images are highly recommended for all figures.
Author Response
"Please see the attachment"

Round 2
Reviewer 1 Report
Comments and Suggestions for Authors
The authors have sufficiently addressed my comments. I recommend that the manuscript be accepted in the present form.